# Effects of Animal Diet and Processing Methods on the Quality Traits of Dry-Cured Ham Produced from Turopolje Pigs

**DOI:** 10.3390/ani14020286

**Published:** 2024-01-17

**Authors:** Danijel Karolyi, Martin Škrlep, Nives Marušić Radovčić, Zoran Luković, Dubravko Škorput, Krešimir Salajpal, Kristina Kljak, Marjeta Čandek-Potokar

**Affiliations:** 1Division of Animal Science, Faculty of Agriculture, University of Zagreb, Svetošimunska cesta 25, 10000 Zagreb, Croatiadskorput@agr.hr (D.Š.); ksalajpal@agr.hr (K.S.); kkljak@agr.hr (K.K.); 2Agricultural Institute of Slovenia, Hacquetova ulica 17, 1000 Ljubljana, Slovenia; martin.skrlep@kis.si (M.Š.); meta.candek-potokar@kis.si (M.Č.-P.); 3Faculty of Food Technology and Biotechnology, University of Zagreb, Pierottijeva 6, 10000 Zagreb, Croatia; nmarusic@pbf.hr

**Keywords:** dry-cured ham, meat smoking, Turopolje pig, acorn

## Abstract

**Simple Summary:**

Innovations in the value chain of traditional meat products, leading to higher quality or healthier products, can support the niche market for local pig breeds and contribute to their more sustainable conservation. In this context, the present study investigated whether the quality traits of smoked dry-cured ham derived from the local Turopolje pig could be improved by including acorns in the animal’s diet or by innovations in processing methods, such as smoke reduction. The results show that feeding acorns to pigs increased the processing yield but had a limited effect on the quality of the dry-cured ham, as only a few differences in the physicochemical, textural or colour parameters and volatile profile were observed. However, some sensory attributes, such as odour typicality, were affected by the acorn diet. Conversely, this innovation in processing led to a significant reduction in smoke-derived volatile compounds and an improved texture to the lightly smoked hams, as shown by both instrumental and sensory analysis.

**Abstract:**

The Turopolje pig (TP) is a local Croatian pig breed that almost became extinct in the second half of the 20th century. Today, the TP is still endangered, and a new conservation strategy based on products with higher added value is needed to preserve the breed. There is little information on the quality of TP meat products such as smoked and dry-cured ham, including the impact of natural feeds or processing innovations such as smoke reduction. This study, therefore, investigated the effects of the animal’s diet (either conventionally fed or acorn-supplemented) and the processing method (standard or lightly smoked) on the quality traits of dry-cured TP ham. Twenty hams, evenly distributed among the treatments, were processed for 15 months and then analysed for physicochemical and textural traits, volatiles and sensory profile. The hams from acorn-supplemented pigs lost less weight during processing (*p* ≤ 0.05). Otherwise, the diet had no significant effect on most examined ham traits. The exceptions were protein content and the texture parameter hardness, which decreased (*p* ≤ 0.05), and the degree of proteolysis and colour parameters, which increased (*p* ≤ 0.05) as a result of acorn supplementation. However, these effects were generally small and varied between the inner (*m. biceps femoris*) and outer (*m. semimembranosus*) muscles. Furthermore, acorn supplementation was associated with less typical ham odour and lower sensory scores for sweetness and colour uniformity (*p* ≤ 0.05). The smoke reduction had no effect on the physicochemical and colour properties but resulted in a significant reduction (*p* ≤ 0.05) in the volatile phenolic compounds and an improved texture to the hams. This was reflected both in reduced (*p* ≤ 0.05) hardness, identified in the instrumental analysis, and in an increased (*p* ≤ 0.05) softness, solubility and moistness, identified in the sensory evaluation. To summarize, the quality of the TP ham under the conditions studied was only slightly affected by acorn supplementation, whereas reduced smoking had a more significant effect, which was mainly reflected in an improved texture.

## 1. Introduction

The Turopolje pig (TP) is a local fatty pig breed that originated in the Turopolje region of central Croatia in the early Middle Ages. Due to its modest requirements, resilience and good adaptation to foraging in the local oak forests and marshy meadows, the TP was an important meat source for centuries, and in the early 1920s it was the most important pig breed in the region, with a population of around 85,000 animals [1]. However, with the transition from extensive to intensive pig farming in the second half of the 20th century, the TP was gradually replaced by imported high-performance pig breeds and crossbreeds. The result was a significant decline in population size, and since 1996, the TP has been under state protection [2]. The conservation process is largely maintained through subsidies to farmers; still, the breed has not yet been targeted for selection or marketing [3]. As a result, despite support, the TP population has been very slow to recover and, with the latest census of 240 sows and 18 boars kept on 18 farms [4], the breed’s existence remains at risk.

The preservation of a breed usually depends on its economic importance. As local breeds are inferior in most of the economically important traits, achieving cost-effective competitiveness and sustainability may require an alternative approach focusing on high-value-added products and niche markets [5,6,7,8]. Indeed, several local pig breeds reared in extensive production systems, for instance, those in traditional silvopastoral systems in southern Europe, are already successfully used for high-value traditional products (e.g., dry-cured meats) that are widely recognised and appreciated [9,10,11]. Moreover, consumer preferences and willingness to pay for such products are increasing across Europe [12,13], opening up market niches for local pig breeds. In addition, consumers today demand not only traditional foods but also healthier types of products [14]. As a result, meat processors have begun to innovate to improve the nutritional and health properties of meat products [15].

The conservation of untapped local pig breeds from different European regions, including the TP, through their better use in traditional production systems and in value-added and innovative products has recently been recognised in the framework of the EU-funded H2020 project, TREASURE [16,17,18,19]. Studies among Croatian consumers revealed that information about the traditional production system, i.e., free-range farming and feeding with acorns, should be emphasized in marketing strategies for the TP breed, as this type of information can influence consumers’ motives for eating and buying TP products such as dry-fermented sausages [3] and dry-cured ham [16,17]. In addition, the health-related innovations in traditional meat products, such as smoke reduction [14], appear to be feasible for better commercialization of this breed, as no significant differences in consumer acceptance were found between the lightly smoked and standard TP dry-cured ham [17]. However, there is little information on the actual quality traits of TP meat products such as smoked and dry-cured ham, including the impact of naturally available feeding sources, e.g., acorns, or health-related processing innovations such as smoke reduction. The aim of the present study was, therefore, to investigate the effect of the diet (acorn supplementation or a conventional diet) and the ham processing method (standard or lightly smoked) on the physicochemical and textural traits and volatile and sensory profile of dry-cured ham originating from the TP breed.

## 2. Materials and Methods

### 2.1. Ham Origin and Processing

The raw hams for this study originated from an observational study conducted in the frame of the H2020 project, TREASURE (Grant Agreement Nº 634476). Twenty TP pigs were kept in an outdoor production system, with a wooden shelter and fenced forest/pasture area (ca. 6000 m^2^), at the experimental facility of the Faculty of Agriculture, University of Zagreb, located in the Turopolje forest. The trial did not involve any procedures with animals that would require approval according to Directive 2010/63/EU for animal experimentation, and the study protocol was approved by the Ethical Committee of the TREASURE project. During the first part of the trial (which started at an average age of 400 days and a live weight of 45 kg), all pigs were housed together and fed 2 kg of commercial feed mixture (ST-II, Fanon d.o.o., Petrijanec, Croatia) per animal, distributed as a single daily ration in the morning. At an average age of 506 days, the pigs were allocated either to the experimental group (*n* = 10), which received a feed mixture supplemented with acorns from the European oak (*Quercus robur*), or to the control group (*n* = 10), which continued to receive only the aforementioned feed mixture. Pigs were kept in two separate enclosures of the same size. During the final fattening phase, which lasted for 46 days, the control pigs received 3 kg of feed mixture per head daily. The acorn-supplemented pigs received 2 kg of feed mixture and were additionally given 2 kg of acorns per head daily. The higher daily amount of feed in the acorn-fed group was intended to compensate for the lower metabolisable energy derived from the acorns compared to the feed mixture. The animals had free access to water. The chemical composition and the composition of the main fatty acid groups of the feed mixture and acorns used in the experiment are shown in Table 1.

At an average age of 552 days and an average live weight of 95 kg, all pigs were transported to a commercial slaughterhouse (ca. 7 km from the experimental site) and slaughtered on the same day, according to the standard abattoir procedure. After 24 h of chilling, the carcass halves from both groups were delivered to a commercial meat processing plant (IGOMAT d.o.o., Otruševec, Croatia) and were separately cut into primary parts, from which the raw hams (10 per feeding treatment) were selected for further processing.

The trimmed thighs were weighed and rubbed by hand with a mixture of curing salt (NaCl with 0.6% NaNO_2_) and spices (black pepper, garlic, and paprika), piled into large PVC tubs and left to be salted in the cold (T = 4 °C, RH = 85) for 5 weeks. After salting, the hams were cold-smoked in a smokehouse (T = 20 °C, RH = 80%) with beech wood smoke a total of 8 times for 60 min and were then transferred to a chamber for drying and ripening under controlled conditions (T = 14 °C, RH = 72%). For the lightly smoked hams, the difference in procedure from the standard processing described above was a 50% reduction in the application of smoke (4 instead of 8 applications). The distribution of hams, grouped according to processing method and the animal’s diet, was a 2 × 2 factorial design, with half of the lightly smoked hams coming from control-fed pigs and the other half coming from acorn-fed pigs, and the same distribution was applied for the standard smoking process (5 hams for each of 4 groups or 10 hams per effect of processing and diet). At the end of processing, when the hams were 15 months old, they were reweighed to determine the processing weight loss and sampled for the planned analyses. The sample sections (ca. 5 cm thick), containing parts of the outer (*m. semimembranosus*—SM) and inner (*m. biceps femoris*—BF) muscles and subcutaneous fat, were excised from the caudal part of the ham, then the samples were coded, vacuum-packed and frozen at −20 °C until analysis. Prior to analysis, the samples were thawed for 24 h at 4 °C.

### 2.2. Physicochemical Analysis

Muscle samples (of BF and SM) were trimmed of superficial connective and fat tissue and pulverized in liquid nitrogen with a laboratory mill (Grindomix GM200, Retsch GmbH and Co., Haan, Germany). The total nitrogen, non-protein nitrogen (NPN) and salt (NaCl) contents were chemically determined, as described by Škrlep et al. [20]. Briefly, the content of NaCl was determined by potentiometric titration using a DL53 general purpose titrator (Mettler Toledo, Schwarzenbach, Switzerland), then the proteolysis index (PI) was calculated as a ratio between the previously determined NPN and the total nitrogen content, while intramuscular fat and moisture contents were determined by near-infrared spectral analysis (NIR Systems 6500 Monochromator, Foss NIR System, Silver Spring, MD, USA) using internal calibrations developed at the Agricultural Institute of Slovenia.

Lipid oxidation was measured by assaying the 2-thiobarbituric acid-reactive substances (TBARS) in the muscle (BF only) using the method employed by Botsoglou et al. with slight modifications [21]. Briefly, 2 g of the sample was weighed in a 50 mL PP test tube, and 10 mL of 5% aqueous trichloroacetic acid (TCA) and 5 mL of 0.8% butylated hydroxytoluene (BHT) in hexane was added to each tube. The contents were homogenized using the T10 basic Ultra Turrax homogenizer (IKA-Werke GmbH & Co. KG, Staufen, Germany) for 30 s at high speed, and then centrifuged for 5 min at 2500× *g* (Centric 322A, Tehtnica Železniki d.o.o., Železniki, Slovenia). The upper hexane layer was discarded and the bottom aqueous layer was filtered (grade 391 filter paper; Munktell & Filtrak GmbH, Bärenstein, Germany). A 2.5 mL aliquot was pipetted into a screw-capped PP tube, then 1.5 mL of 0.8% aqueous 2-thiobarbituric acid (TBA) was added and the contents were incubated for 30 min at 95 °C (Memmert GmbH, Schwabach, Germany). After incubation, the tubes were cooled under tap water, and their absorbance at 532 nm was determined (Helios γ, Thermo Electron Ltd., Rugby, UK) against a blank containing 2.5 mL of TCA and 1.5 mL of 0.8% aqueous TBA. The content of malondialdehyde (MDA) was calculated from the standard curve using 1,1,3,3-tetrametoxypropane. The TBARS were measured in triplicate and were expressed as mg of MDA per kg of tissue.

For pH measurement, a suspension of 10 g of muscle sample was mixed with 90 mL of distilled water, and the pH was measured with a pH meter (TESTO 230, Testo SE & Co., KGaA, Titisee-Neustadt, Germany). Two-point calibration with the Testo buffer pH 4/7 set was performed before the measurements were taken, according to the manufacturer’s instructions. Water activity was measured with the HygroPalm AW1 SET instrument (Rotronic AG, Bassersdorf, Switzerland), using the Aw Quick mode, in sub-samples taken after coarse homogenisation of 80 g of the muscle. The Aw and pH analyses were performed in triplicate.

### 2.3. Colour Measurements

The objective colour parameters (CIE L*, a*, b*) of the BF and SM muscles and of the subcutaneous fat were measured using a Minolta Chroma Meter CR-300 (Minolta Co., Ltd., Osaka, Japan) with an 8 mm measuring area. The CIE standard illuminant D_65_ was used as the light source for the measurement. Before measurement, calibration was carried out on the CR-A43 white calibration plate in D_65_, where Y = 93.5, x = 0.3156, and y = 0.3319. From the measured chromatic coordinates a* and b*, the Chroma (C*) value was calculated as √(a*2 + b*2), and the Hue angle (h°) value as tan^−1^(b*/a*), indicating colour saturation and taint, respectively. Colour measurements were made in quadruplicate.

### 2.4. Texture Profile Measurements

Instrumental texture profile analysis (TPA) of the BF and SM muscles was performed as described by Pugliese et al. [22], using a texture analyser (Ametek Lloyd Instruments, Ltd., Bognor Regis, UK) with a 50 kg load cell and a compression plate of 50 mm in diameter. Prior to analysis, the muscles were excised with a scalpel, trimmed of fat and connective tissue, and carefully cut into parallelepipeds measuring 20 mm × 20 mm × 15 mm (length × width × height). The samples were compressed twice, down to 50% of their original height, perpendicular to the fibre bundle and at a crosshead speed of 1 mm/s, and the following parameters were calculated from the force-distance curves: hardness (N), cohesiveness, gumminess (N), springiness (mm), chewiness (N×mm) and adhesiveness (N×mm). All the TPA measurements were performed in duplicate.

### 2.5. Analysis of Volatile Compounds

For the analysis of the volatile profile, a subset of 8 dry-cured hams (2 for each of the 4 groups, with 4 per effect of processing and diet) was examined. Volatile organic compounds (VOCs) were determined, based on the method used by Petričević et al. [23]. The extraction of headspace VOCs was performed using a solid phase microextraction (SPME) device containing a fused silica fibre (20 mm length) coated with a 50/30 μm DVB/Carboxen/PDMS fibre (Supelco, Bellefonte, PA, USA). Muscle samples (BF only) were homogenized in distilled water that was saturated with NaCl, using a commercial blender. Ten mL of this mixture was placed into each 20 mL vial, 100 μL of 4-methyl-2-pentanol (1.2 mg/kg) (internal standard) was added and the vials were tightly capped with a PTFE septum. Duplicate 20 mL vials were placed in a thermoblock at 40 °C. The SPME fibre was then exposed to the headspace for 180 min while maintaining the sample at 40 °C. After extraction, the SPME fibre was immediately injected into a 6890 N gas chromatograph coupled to a 5975 series mass selective detector (Agilent Technologies, Santa Clara, CA, USA). A capillary column DB-5 MS 30 m × 0.25 mm, film thickness 0.25 μm (Agilent Technologies, Santa Clara, CA, USA), was used with helium as a carrier gas at a 1.0 mL/min flow rate. The temperature of the injector, used in the splitless mode, was 230 °C and the desorption time was 5 min. The temperature programme was set at 40 °C, then isothermal for 10 min, then rising to 200 °C at a rate of 5 °C/min and finally rising to 250 °C at a rate of 20 °C/min. The final temperature was held for 5 min. The transfer line temperature was maintained at 280 °C. The mass spectra were obtained at 70 eV with a rate of 1 scan/s over an *m*/*z* range of 50–450. An in-house mixture of C8–C20 n-alkanes was run under the same chromatographic conditions to calculate the retention indices (RI) of the detected compounds. AMDIS 3.2, program version 2.62, was used for the identification of components, using the NIST 2005 version 2.0 spectral library (NIST, Gaithersburg, MD, USA) as well as for a comparison of the obtained RI with values given in the literature ([24] and the in-house library).

### 2.6. Sensory Analysis

The sensory evaluation was carried out by means of quantitative descriptive analysis [25] using a sensory panel of 9 trained members. Before the first evaluation, a “calibration” session was held with a variety of dry-cured hams, differing in, e.g., the degree of dryness, saltiness, softness, smoking, etc., during which the panel members decided on the sensory descriptors and evaluation scale. A total of 19 sensory descriptors were evaluated: 5 described the visual attributes of the whole slice (marbling, colour of the muscle tissue, colour uniformity, colour intensity and colour of the subcutaneous fat), 2 described the odour of the whole slice (typicality and smoke intensity), 3 described the taste/flavour of the subcutaneous fat (rancid, sweet and off-flavours), 5 considered the taste/flavour of the BF muscle (saltiness, sourness, sweetness, bitterness and off-flavour) and 4 described the texture of the BF muscle (softness, solubility, juiciness and pastiness). The intensity of the descriptors was measured on a 9 cm linear, unstructured scale anchored at both extremes, from “not perceived” (left) to “very intense” (right). Individual samples of dry-cured ham (1 slice, 1 mm thick) were placed on white plastic trays coded with 3-digit numbers. The samples were presented to the evaluators in a random order at room temperature. Fresh water and a piece of bread with slices of apple were used as a taste neutraliser between the samples. A total of 5 assessment sessions were conducted, with 4 samples (1 per treatment) tested in each session.

### 2.7. Statistical Analysis

Data on the ham processing losses, physicochemical and colour traits, instrumental texture measurements and VOCs were analysed by a two-way analysis of variance with the fixed effects of diet (D) and processing (P), and their interactions were assessed using the general linear model (GLM) of the SAS 9.4 statistical software (SAS Institute Inc., Cary, NC, USA). Since there was no interaction between the processing method and diet, the single effects are reported below, with the exception of VOCs. For sensory traits, a repeated measures analysis (using a panellist as a random effect) was conducted using the MIXED procedure in SAS 9.4. To assess the differences between treatments, the least squares means (LSM) were compared, using Tukey’s *t*-test at a significance level of 5%. Effects tending towards statistical significance (*p* < 0.10) are labelled with t in the tables. Due to the small sample size for the volatile profile, the effect sizes for all identified VOCs were calculated as Hedge’s *g* value.

## 3. Results

### 3.1. Processing Weight Loss, Physiochemical and Colour Properties of TP Dry-Cured Ham

The results for processing loss, physicochemical properties and the colour of BF and SM muscles, as well as the colour of the subcutaneous fat of TP dry-cured hams, are shown in Table 2.

Compared to the control hams (CH), the hams from acorn-fed pigs (AH) had lost less weight and were heavier at the end of processing, whereas the processing method itself had no effect (*p* > 0.05) on the weight and processing yield of the hams.

In terms of physicochemical traits, the AH hams had a lower protein content and a higher proteolysis index (ratio of non-protein to total nitrogen) and tended (*p* < 1.0) to contain more moisture in the BF muscle. In the SM muscle, there were only tendencies towards higher moisture and lower protein and fat contents in the AH compared to the CH. The salt content, the water activity (a_w_), the pH values and the TBARS (BF only) were not influenced (*p* > 0.05) by the pigs’ diet.

The effect of acorn feeding on the colour measurements showed higher lightness (L*), redness (a*), yellowness (b*) and chroma (C*) values in the SM but not in the BF muscle, with the exception of hue (h°), which was higher in both muscles compared to the control. For the subcutaneous fat, the a* and C* values were increased and the L* value tended (*p* < 1.0) to be higher in the AH.

The effects of processing methods on physicochemical traits, lipid oxidation and colour properties were generally not significant (*p* > 0.05), with only a tendency (*p* < 1.0) towards lower protein content in the lightly smoked hams (LSH) than in the standard smoked hams (SH).

### 3.2. Instrumental Texture Profile of TP Dry-Cured Ham

The results for the TPA of TP dry-cured ham in relation to the animal’s diet and the processing method are shown in Table 3.

The results show that most of the texture profile traits were not significantly affected (*p* > 0.05) by the diet or processing method, regardless of the muscle. An exception was hardness (N), a texture parameter that was only significantly lower for the BF muscle in both the AH and the LSH, compared to the CH and SH, respectively. In addition, the cohesiveness tended (*p* < 1.0) to increase in the SM muscle and the gumminess tended (*p* < 1.0) to decrease in the BF muscle with acorn feeding and smoke reduction, respectively.

### 3.3. Volatile Compounds of TP Dry-Cured Ham

The effects of the animal’s diet, the processing method and their interaction on the VOCs of TP dry-cured ham BF muscle are shown in Table 4 and Appendix A.

A total of ninety-one (**91**) VOCs were identified (Appendix A), of which aldehydes were the most abundant and with the highest number of volatile compounds (**16**), followed by phenols (**14**), alcohols and ketones (**13**), alkanes and alkenes (**12**), aromatic hydrocarbons (**10**), terpenes (**8**), nitrogen compounds (**3**) and acids (**2**).

Overall, the processing method (i.e., smoking) had a more marked influence on the volatile profile of the dry-cured ham than the diet of the animals. As expected, the LSH contained a significantly lower amount of total phenolic compounds, mainly due to the lower contents of 3-ethyl-phenol, 2-methoxy-4-methylphenol and 4-ethyl-2-methoxy-phenol compared to those in the SH. The effect of smoking duration on the other groups of VOCs was less obvious, with differences found only for some individual compounds within certain groups of volatiles. Of the aldehydes, the LSH had lower levels of tetra-decanal and hexa-decanal and tended (*p* < 0.10) to have lower levels of nonanal, which, along with hexanal and heptanal, was the most abundant aldehyde and the only major aldehyde where the level was inclined to be reduced by a reduction in smoking. The tendency (*p* < 0.10) to decrease due to smoking reduction was also observed for 1,2,3-trimethoxybenzene within the group of aromatic hydrocarbons. In the group of alkanes and alkenes, the LSH contained less cyclododecane and tended (*p* < 0.10) to have a lower content of pentadecene and 1-pentadecene than the standard hams, while the content of 3-methyl-heneicosane was higher. Furthermore, the LSH had more alcohol 4-methyl-1-(1-methylethyl)-3-cyclohexenol and tended (*p* < 0.10) to have higher levels of total terpenes, ketone 1-octen-3-one and N-compound 2,6-dimethylpyrazine.

The influence of diet on the VOCs of dry-cured ham was generally insignificant, except in the group of aldehydes, where tetradecanal was lower, while nonanal and 2-methylbutanal tended (*p* < 0.10) to be higher in “acorn-fed” hams compared to the control. In addition, several significant (*p* ≤ 0.05) interactions between diet and processing method on ham volatiles were observed (Appendix A). For instance, differences between processing methods appeared to be diet-dependent for some ketone compounds, with higher levels of 2-decanone in the LSH than in the SH for only the control diet, while 6,10-dimethyl-5,9-undecadien-2-one was higher in the SH but only for the acorn diet. In addition, the total acids were significantly higher in the LSH of the acorn-fed group than in the SH for both diets, while in the control-fed group, the acid levels between the smoking processes were similar and were comparable to those observed in their acorn diet counterparts. Similar interactions between diet and processing were also observed for a few of the less well-represented VOCs within the phenolic compounds and alkenes (Appendix A).

Hedge’s *g* effect sizes regarding an animal’s diet and processing method on the VOCs of TP dry-cured ham BF muscle are shown in Figure 1 and Appendix A.

The effect size for diet was small to medium for all VOC groups. However, for some of the individual VOCs, there was a large difference in the number of standard deviations by which the means of the AH and CH differed (Hedge’s *g* > 0.8; Appendix A), indicating a possibly large effect size for diet. This was particularly evident for several aldehydes, including nonanal, which appeared to be increased by the acorn diet, as already shown in the analysis of variance. Looking at the magnitude of effect size for processing, a large positive Hedge’s *g* value for total phenols, aromatic hydrocarbons and nitrogen compounds indicates lower mean values of these VOCs in the LSH group than in the SH group. On the other hand, alcohols, ketones, terpenes and acids show a large negative Hedge’s *g* value overall, which indicates sizably higher group means in the LSH group than in the SH group. In addition, either positive or negative large effect sizes for processing were found for many individual VOCs (Appendix A). However, with a few exceptions (mainly for phenols), most of the observed effect sizes were not confirmed by the analysis of variance, which is probably due to the small sample size that was available for analysing the VOCs.

### 3.4. Sensory Analysis of TP Dry-Cured Ham

The effects of the animal’s diet and the processing method on the sensory traits of TP dry-cured ham are shown in Table 5.

Regarding the sensory evaluation of the whole ham slice, marbling, fat colour and the intensity of meat colour, as well as smoke intensity, were comparable between the diets (*p* > 0.05). However, the uniformity of the meat colour was rated as significantly lower in the AH, while the meat colour tended (*p* < 0.1) to be rated lower compared to that of the CH. The typical odour of the whole ham slice also seemed to be affected by the animal’s diet, as the AH samples were perceived by the sensory panel as having a less typical odour than hams from animals fed the control diet. Conversely, a reduction in smoking had no significant effect (*p* > 0.05) on the perceived odour and visual attributes of the whole ham slice, with the exception of a tendency (*p* < 0.1) towards lower meat color intensity in the LSH.

In terms of taste and flavor attributes, the rancidity and sweetness of the fat part and the saltiness of the muscle part did not differ (*p* > 0.05) between the hams from different diets or smoking processes. However, compared to the CH, the AH samples were rated as less sweet in the BF muscle part, with a tendency (*p* < 0.1) towards a low but slightly higher perceived off-flavour in both the fat and BF muscle, none of which was affected by smoking. In addition, diet had no effect on the sourness or bitterness of the BF, while a reduction in smoking tended (*p* < 0.1) to reduce this attribute in the LSH.

Finally, a significant effect of smoking, but not of diet, was found in the sensory profiling of the texture attributes of dry-cured ham. The BF muscle of the LSH was perceived as softer, more moist and more soluble than in the standard product, while the reduction in smoking had no effect (*p* > 0.05) on the pastiness of the cured BF muscle.

## 4. Discussion

Feeding starch-rich tree nuts such as acorns to pigs from breeds with a low potential for muscle growth [26], such as TP, leads to the rapid development of adipose tissue and the deposition of large amounts of lipids under the skin, situated in and/or between the muscles [27,28,29]. The properties of free-range pork are also enhanced because many unique compounds such as FAs and antioxidants, derived from acorns, grass and other natural feedstuffs that are directly foraged by the pigs in the woodlands and meadows, are absorbed and stored in the adipose and other tissues [30,31,32,33,34]. The quantity and quality of these fat deposits are considered essential for the increased suitability of pork for dry-curing processing [11] and for the final quality of the product, in terms of its analytical and sensory properties. As a result, separate compositional and sensorial properties [35,36,37,38], including the volatile [37,39,40] and FA profiles [38,40] of dry-cured meat products from free-range acorn-fed pigs have been reported.

In the present study, feeding acorns to pigs in the finishing stage (2 kg per head per day) as part of a mixed diet resulted in lower weight loss during ham processing and a higher yield of dry-cured product, as well as lower protein content and a higher proteolysis index in the BF muscle of “acorn-fed” hams. This result could be due to the lower protein content (2.5 vs. 15% in acorns vs. the control cereal-based feed, respectively) and high starch content of acorns (20.7%), which may favour the deposition and increase in thickness of fat tissue [41], thus reducing dehydration losses during ham processing. Indeed, the backfat thickness of the acorn-fed pigs in this experiment was higher than that of the control group (on average 43.1 ± 6.8 mm vs. 38.6 ± 5.9 mm). However, the intramuscular fat content, a trait that has been positively related to product juiciness and consumers’ preferences [42], was not increased by the acorn diet. The lower processing losses for the AH can be related to the higher moisture content, the latter also explaining the higher proteolysis [43] observed in the BF muscle. This could also explain the slightly lower hardness (N) of the BF of the AH hams, although, generally, no significant differences in the water activity and other physicochemical or textural characteristics of the hams were observed in association with the diet [43].

The effect of acorn feeding on the colour parameters of dry-cured ham was inconsistent: the hue angle (h°) increased in both the BF and SM muscles, while lightness (L*), redness (a*), yellowness (b*) and chroma (C*) only increased in the SM muscle. In terms of subcutaneous fat a* and C*, these were also increased in association with the acorn diet. These results are partly in agreement with those of Pugliese et al. [44], who reported that free-range rearing on oak and chestnut woodland pastures resulted in higher redness of muscle tissue in dry-cured hams from *Cinta senese* pigs, compared to indoor rearing and commercial feeding. In another study on Iberian hams [45], no effects of the finishing diets (acorn vs. commercial feed) were reported on ham muscle colour. The diet influenced the subcutaneous fat colour parameters L*, a* and C*, but, in opposition to the present research, higher values were reported in association with commercial feeding. Obviously, any literature results are difficult to compare with those of the present study, as the colour of meat products such as dry-cured ham also depends on various factors other than diet, such as the genotype and age of the animals [46], muscle pigment concentration and chemical state [47], fat content and FA composition [48], or processing [49]. It should be noted, however, that the dry-cured musculature of the TP ham analyzed is much redder and less pale than that of commercial dry-cured and smoked Croatian hams from conventional pig production [50].

The changes in the composition of pig tissues under acorn supplementation and the distinct profile of the volatile compounds that develop during ham processing [39,40] are considered crucial for the characteristic aromatic and sensory properties of dry-cured products. This was also evidenced by the higher consumer preferences for “acorn-fed” dry-cured hams from traditional silvopastoral production systems, based on acorn supplementation [51]. For example, *Montanera* hams from Iberian pigs fattened outdoors exclusively on acorns and pasture had higher concentrations of aldehydes and ketones and lower concentrations of sulphur-containing compounds than *Campo* hams from Iberian pigs fattened on concentrate with a high oleic acid content [39], while Tuscan dry-cured ham from *Cinta senese* pigs fed acorns under free-range conditions had the highest levels of aldehydes and esters compared to hams from their counterpart pigs fed chestnuts or confined pigs fed commercial feed [40]. In the present study, the effect of acorn feeding on the VOCs of dry-cured ham was generally insignificant, with only nonanal (the second most abundant aldehyde after hexanal) exhibiting increased values in association with acorn supplementation. Aldehydes are known to contribute most significantly to the unique flavour of cured ham, due to their low threshold and abundance [52]. Nonanal, a saturated aldehyde associated with the autoxidation of unsaturated fatty acids such as oleic, linoleic, linolenic and arachidonic acids [53], has been shown to contribute to green, greasy, rancid flavours and a rancid odour [54]. This could be the reason why the sensory panel rated the odour of AH as being less typical. In addition, the possible unfamiliarity of the evaluators with the sensory characteristics of meat products from acorn-fed pigs, which are new to the local market, should also be considered. Apart from this finding, the less homogeneous slice colour and the slightly lower BF sweetness, the panel could not perceive any other diet-associated sensory differences. The lack of the acorn diet’s effect on the volatile and sensory profile in the present study could also be due to the different amounts and/or types of acorns used. Namely, acorns from the Mediterranean oaks, *Q. ilex*, *Q. rotundifolia* and *Q. suber* are characterized by a distinct phenolic and FA composition (e.g., >63% of oleic acid in the total FA) [55]. Moreover, the quantity of acorns consumed by pigs, as part of a mixed diet, in the present study does not seem to be sufficient for appreciable changes in the aroma and sensory characteristics of the dry-cured hams.

Regarding the impact of the ham processing method, the present study examined smoke reduction, which has recently been recognised by Croatian consumers as the most widely accepted health-related innovation in traditional meat products with the least negative impact on the perceived traditional character of the product [14]. Unlike most dry-cured hams from other Mediterranean countries, hams in Croatia are usually smoked using hardwood. Smoking gives dry-cured meats the desired sensory characteristics, such as a golden-brown surface colour and the typical flavour and aroma of smoked meat, which are highly preferred by local consumers. When meat is smoked, the desired colour is formed when compounds with a carbonyl group (aldehydes and ketones) are absorbed from the smoke into the moist surface of the meat, reacting with amines from the protein, while the typical flavor and the formation of a “smoky” aroma occur mainly due to the absorption of phenolic compounds that are present in smoke and their subsequent diffusion into the deeper portions of the product [56]. In addition, the smoke possesses antimicrobial [57] and antioxidant effects [58] and acts as a barrier against the development of rancidity due to the protective film it creates on the dry-cured ham’s surface [59]. Conversely, smoking meat can cause potentially harmful compounds from the smoke, such as polycyclic aromatic hydrocarbons (PAHs), to be deposited in the products, but the extent of PAH contamination can be greatly reduced by controlling the intensity and conditions of the smoking process [60].

In the present study, smoking was carried out under controlled conditions in the smokehouse and the amount of smoke used was reduced by half in the LSH group. This reduced smoking had no negative effect on the physicochemical and colour properties of the ham but seemed to improve the texture profile, as evidenced by the lower hardness of the BF in the LSH. Furthermore, intramuscular lipid oxidation was not affected by the reduced smoking. In fact, in all the analysed hams, the TBARS levels were quite low (~0.70 mg MDA/kg), indicating a possible underestimation of MDA levels that may occur in meats cured with nitrite curing salt, due to the nitridation of MDA blocking the reaction with TBA [61]. Conversely, the higher intake of potent antioxidants such as α-tocopherol and carotenes [55,61] from acorns, grass and other natural feedstuffs that are available to free-range pigs may also contribute to the higher oxidative stability of the hams [35]. Regarding the antimicrobial effect of the smoking process, no microbiological analyses were performed in the present study. However, we found no significant differences in the physicochemical parameters such as a_w_ and pH values between the LSH and SH, which could, at least indirectly, indicate similar microbial control in both process types.

As for the volatile profile, phenols originating from smoke were the second most abundant VOCs after aldehydes. Phenols and methoxyphenols are of great importance to the creation of the smoke flavour, due to their woody, pungent, burnt, ashy, petroleum-like and smoky notes, which can already be sensed at low concentrations of 1–10 mg/kg [62]. Hence, their presence is a unique characteristic of smoked and dry-cured hams such as Croatian *Dalmatinski pršut* and *Drniški pršut* [23,59,63] or other smoke-cured meat products [64]. As expected, the reduction in smoking resulted in a significant decrease in smoke-derived phenolic compounds, which was the most significant difference in the volatile profiles established between the smoking processes. However, this was not detected by sensory profiling, as the panellists responded similarly to the “smoke intensity” attribute of all hams, which is likely due to the low threshold of the phenolic compounds present [50]. The contents of phenols in this study were generally lower than in dry-cured hams that were smoked using the traditional smoking method, without a smoke chamber [50], which may also diminish the difference in perceived smoke intensity between the lightly smoked and standard hams.

Conversely, the sensory profiling corroborates the results of the texture profile analysis of the BF muscle, which was assessed by the panellists as softer, moister and more soluble in lightly smoked hams, implying the beneficial effect of reduced smoking on the perceived texture of the dry-cured ham.

Finally, it should be pointed out that the present study has potential limitations, due to the small sample size that was analysed. This was particularly evident in the profile of the volatiles, where large effect sizes (Hedge’s *g* > 0.8) indicate the different mean values of several individual and total VOCs between the experimental and control hams; however, this was mostly not confirmed by the analysis of variance due to the small sample size. Therefore, further research with a larger sample size is needed to confirm the observed effects.

## 5. Conclusions

The present study has demonstrated that supplementing the diet of TP pigs in the finishing phase with acorns is associated with higher ham processing yield. Only a few differences in the physicochemical, textural or colour parameters and the volatile profile were observed, suggesting that a higher acorn intake (e.g., >2 kg per head daily) might be required. However, some of the sensory attributes of the ham, such as odour typicality, seem to be altered by an acorn diet, which should be taken into account when planning similar studies. Conversely, reducing the smoking of ham by 50% proved to be an effective innovation in processing that can improve the texture of dry-cured ham without causing major changes in other quality traits.

## Figures and Tables

**Figure 1 animals-14-00286-f001:**
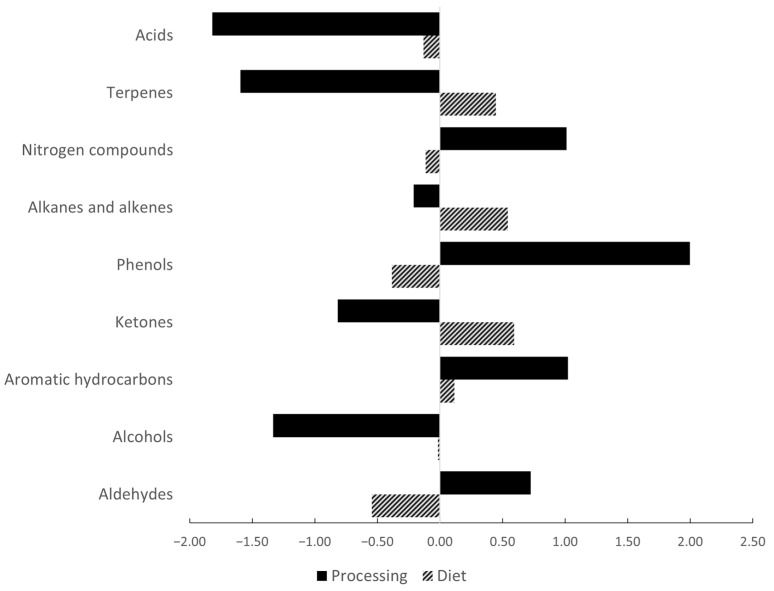
Effect sizes (Hedge’s *g*) of diet and processing on the total amounts of each VOC group of TP dry-cured ham *m. biceps femoris* muscle, denoting the differences between the experimental and control groups (i.e., acorn supplementation vs. feed mixture and smoke reduction vs. standard smoking, respectively). Positive values mean a higher result and negative values mean a lower result in the control group than in the experimental group. If Hedge’s *g* is >0.8 of standard deviation, the effect is considered large.

**Table 1 animals-14-00286-t001:** Chemical and fatty acid composition of the feeds.

	Mixture	Acorn
Chemical composition (g/kg):		
Dry matter	891	604
Crude protein	150.2	25.1
Ether extract	30	20
Crude fibre	46	65
Ash	51	13
Starch	-	207.4
Fatty acid composition (%) ^a^:		
Total SFA	15.51	27.89
Total MUFA	30.93	34.58
Total PUFA	53.56	37.53
ME (MJ/kg of DM)	12.96	7.70

^a^ Percentage of total fatty acids quantified; SFA—saturated fatty acids (FA), MUFA—monounsaturated fatty acids, PUFA—polyunsaturated fatty acids; ME—metabolizable energy.

**Table 2 animals-14-00286-t002:** Effects of the animal’s diet and processing method on processing loss and the physiochemical and colour traits of TP dry-cured ham.

	Processing (P)		Diet (D)		RMSE	Significance	
Trait	Less Smoke	Standard	Acorn	Control		P	D
Green ham weight, kg	6.7	6.5	6.9	6.3	0.86	ns	ns
Ripe ham weight, kg	4.5	4.1	4.6	4.0	0.56	ns	*
Processing loss,%	33.9	34.4	33.1	35.3	2.0	ns	*
*M. biceps femoris*							
Moisture, g/kg	557.2	548.6	558.8	547.0	13.6	ns	t
Fat, g/kg	66.9	61.3	65.5	62.7	11.1	ns	ns
Protein, g/kg	256.0	267.0	253.0	270.1	12.5	t	*
Salt, g/kg	74.3	78.7	77.0	76.0	7.0	ns	ns
a_w_	0.894	0.890	0.894	0.889	0.02	ns	ns
pH	6.00	6.00	5.96	6.04	0.13	ns	ns
Proteolysis index, %	12.2	10.9	12.6	10.5	1.8	ns	*
Lightness, L*	36.8	35.7	36.4	36.0	1.8	ns	ns
Redness, a*	15.8	16.0	15.9	16.0	0.84	ns	ns
Yellowness, b*	5.5	5.5	5.6	5.3	0.50	ns	ns
Chroma, C*	16.7	16.9	16.8	16.9	0.93	ns	ns
Hue, h°	19.0	18.8	19.5	18.3	1.1	ns	*
TBARS, mg MDA/kg	0.71	0.68	0.64	0.75	0.29	ns	ns
*M. semimembranosus*							
Moisture, g/kg	402.5	389.3	411.6	380.2	37.0	ns	t
Fat, g/kg	65.4	66.4	62.7	69.1	7.5	ns	t
Protein, g/kg	451.1	462.0	440.8	472.4	36.9	ns	t
Salt, g/kg	55.6	58.0	58.2	55.4	6.1	ns	ns
a_w_	0.896	0.888	0.895	0.889	0.02	ns	ns
pH	6.08	6.05	6.05	6.08	0.11	ns	ns
Proteolysis index, %	7.3	7.6	7.9	7.0	1.2	ns	ns
Lightness, L*	28.8	28.3	29.1	28.0	1.1	ns	*
Redness, a*	10.6	10.3	11.2	9.7	1.1	ns	*
Yellowness, b*	3.1	3.2	3.6	2.7	0.54	ns	*
Chroma, C*	11.0	10.7	11.7	10.0	1.2	ns	*
Hue, h°	16.3	17.0	17.8	15.6	1.5	ns	*
Subcutaneous fat							
Lightness, L*	72.7	73.7	74.2	72.3	2.3	ns	t
Redness, a*	5.0	4.2	5.2	4.0	1.1	ns	*
Yellowness, b*	3.7	4.3	4.3	3.8	1.2	ns	ns
Chroma, C*	6.3	6.2	6.8	5.7	0.87	ns	*
Hue, h°	37.5	44.8	39.2	43.2	12.7	ns	ns

* *p* ≤ 0.05; t—*p* < 0.10; ns—*p* > 0.05; RMSE—root mean square error; a_w_—water activity; TBARS—thiobarbituric acid-reactive substances; MDA—malondialdehyde.

**Table 3 animals-14-00286-t003:** Effect of the animal’s diet and the processing method on the texture profile of TP dry-cured ham.

	Processing (P)		Diet (D)		RMSE	Significance	
Trait	Less Smoke	Standard	Acorn	Control		P	D
*M. biceps femoris*							
Hardness, N	30.9	44.0	31.7	43.2	11.6	*	*
Cohesiveness	0.49	0.52	0.50	0.52	0.11	ns	ns
Gumminess, N	15.9	24.6	16.4	24.0	10.7	t	ns
Springiness, mm	3.6	3.7	3.7	3.6	0.37	ns	ns
Chewiness, N×mm	58.3	94.8	61.8	91.3	51.1	ns	ns
Adhesiveness, N×mm	−2.1	−1.8	−2.0	−2.0	0.38	ns	ns
*M. semimembranosus*							
Hardness, N	184.4	188.9	163.0	210.4	65.6	ns	ns
Cohesiveness	0.37	0.39	0.40	0.36	0.04	ns	t
Gumminess, N	65.3	71.8	63.0	74.1	22.7	ns	ns
Springiness, mm	3.3	3.3	3.5	3.2	0.74	ns	ns
Chewiness, N×mm	214.6	273.8	254.4	234.0	127.9	ns	ns
Adhesiveness, N×mm	−0.77	−1.0	−1.1	−0.66	0.77	ns	ns

* *p* ≤ 0.05; t—*p* < 0.10; ns—*p* > 0.05; RMSE—root mean square error.

**Table 4 animals-14-00286-t004:** Effect of the animal’s diet and the processing method on the total amounts of the individual VOC groups of the *m. biceps femoris* muscle of TP dry-cured ham.

	Processing (P)		Diet (D)		RMSE	Significance		
Volatile Compounds	Less Smoke	Standard	Acorn	Control		P	D	Interaction
Aldehydes	29.00	35.40	36.61	27.79	7.00	ns	ns	ns
Alcohols	17.06	9.66	13.41	13.31	5.74	ns	ns	ns
Aromatic hydrocarbons	2.81	6.26	2.29	4.78	3.53	ns	ns	ns
Ketones	4.92	3.10	3.27	4.75	1.90	ns	ns	ns
Phenols	9.61	16.17	14.32	11.45	2.77	*	ns	ns
Alkanes and alkenes	5.73	5.27	4.96	6.05	2.18	ns	ns	ns
Nitrogen compounds	4.60	10.91	8.19	7.33	6.57	ns	ns	ns
Terpenes	6.66	1.46	3.50	5.00	2.92	t	ns	ns
Acids	0.40	0.07	0.28	0.19	0.10	*	ns	*

* *p* ≤ 0.05; t—*p* < 0.10; ns—*p* > 0.05; RMSE—root mean square error.

**Table 5 animals-14-00286-t005:** Effect of the animal’s diet and the processing method on the sensory traits of TP dry-cured ham.

	Processing (P)		Diet (D)			Significance	
Trait	Less Smoke	Standard	Acorn	Control	RMSE	P	D
Visual attributes (whole slice)							
Marbling	4.6	4.7	4.7	4.6	1.40	ns	ns
Meat colour	6.3	6.1	6.0	6.5	1.23	ns	t
Meat colour uniformity	5.4	5.0	4.6	5.8	1.56	ns	*
Meat colour intensity	5.8	6.6	6.1	6.3	1.15	t	ns
Fat colour	7.5	7.1	7.2	7.3	1.36	ns	ns
Odour attributes (whole slice)							
Smoke intensity	3.3	3.4	3.4	3.3	2.01	ns	ns
Typicality	6.7	6.4	6.0	7.1	1.31	ns	*
Taste and flavour attributes (subcutaneous fat)							
Rancidity	0.77	1.3	1.3	0.79	1.46	ns	ns
Sweetness	5.7	5.5	5.5	5.7	1.68	ns	ns
Off-flavours	0.56	0.62	0.88	0.30	1.12	ns	t
Taste and flavour attributes (*m. biceps femoris*)							
Saltiness	6.6	6.9	6.8	6.6	1.08	ns	ns
Sourness	3.3	3.7	3.6	3.3	2.21	t	ns
Sweetness	3.4	3.3	3.1	3.6	2.07	ns	*
Bitterness	0.29	0.43	0.38	0.34	0.45	t	ns
Off-flavours	0.43	0.59	0.75	0.27	1.11	ns	t
Texture attributes (*m. biceps femoris*)							
Softness	6.8	6.0	6.5	6.4	1.27	*	ns
Solubility	6.0	5.2	5.6	5.6	1.59	*	ns
Moistness	6.1	5.4	5.8	5.7	1.33	*	ns
Pastiness	1.3	1.4	1.3	1.3	1.52	ns	ns

* *p* ≤ 0.05; t—*p* < 0.10; ns—*p* > 0.05; RMSE—root mean square error.

## Data Availability

The data that support the findings of this study are available from the corresponding author, D.K., upon reasonable request.

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
