# Peer review of "Effects of Animal Diet and Processing Methods on the Quality Traits of Dry-Cured Ham Produced from Turopolje Pigs"

_animals, 2024, doi:10.3390/ani14020286_

Round 1

Reviewer 1 Report

Comments and Suggestions for Authors

A generally well written manuscript with sound experimental design.

The research results presented in the manuscript complement the existing knowledge about the quality of traditional meat products, in this case smoked dry-cured ham from the local Turopolje pig taking into account the animals’ diet and innovations in processing methods, such as smoke reduction.  It provides the reader with an overview of dry-cured meats quality by comprehensively analyze the variation of metabolites during processing and thus to acquire a better understanding of the taste formation process of dry-cured beef. Much of the research can be directly related to worldwide dry-cured meats production so is therefore valuable. Due to the fact that English is not my native language, I am not able to check the grammatical correctness of the manuscript.

The size for an experimental group is probably small but still yields useful data. However, I have a few questions about this section of the manuscript.

1.     Please indicate how many hams were used as experimental material in each of the four experimental groups. As can be seen from the description in the Material&Methods chapter, there were 10 animals in each group of pigs with different diets (line 100-101). 10 hams were obtained from each of both groups (line 116). How was the next step? These 10 hams were divided into two parts (n=5?), one of which was subjected to a normal smoking procedure and the other with a 50% reduction in application of smoke? Please specify.

2.      Why were the samples frozen, since dry-cured hams are usually shelf-stable products? If this way possible biochemical changes were protected? How long were the hams stored frozen? Did the Authors take into account the fact that frozen storage may affect the mechanical properties, sensory quality of the product and initiate fat oxidation?

3.     How were samples cut for texture analysis using the TPA test? During the measurement, was the force directed perpendicular or parallel to the muscle fibers? Depending on the position of the test in relation to the compressing plate (or probe), we talk about a greater impact of the connective tissue (when the muscle fibers are arranged parallel to the base and perpendicular to the force) or a greater impact of the muscle fibers.

4.      Why did the Authors decide to compress the sample to 50% of its original height? Wouldn't it be better to have a destructive effect, i.e. 70-80%, in the case of products with significant gumminess and varying hardness? A much better correlation with the sensory evaluation is obtained when the sample is subjected to disintegration. Similar to what happens when it is crushed in the mouth.

Author Response

Dear reviewer, We would like to thank you for your useful comments and suggestions for improving the manuscript. Below you will find our responses to your comments. All changes made have been incorporated into the revised manuscript by using the “Track Changes” tool for Word. Best regards yours, Danijel Karolyi Corresponding author

Reviewer 1

A generally well written manuscript with sound experimental design.

Thank you very much for your positive general opinion of the paper.

The research results presented in the manuscript complement the existing knowledge about the quality of traditional meat products, in this case smoked dry-cured ham from the local Turopolje pig taking into account the animals’ diet and innovations in processing methods, such as smoke reduction.  It provides the reader with an overview of dry-cured meats quality by comprehensively analyze the variation of metabolites during processing and thus to acquire a better understanding of the taste formation process of dry-cured beef. Much of the research can be directly related to worldwide dry-cured meats production so is therefore valuable. Due to the fact that English is not my native language, I am not able to check the grammatical correctness of the manuscript.

We thank you. We appreciate that you recognise the potentially wider significance of our work.

The size for an experimental group is probably small but still yields useful data. However, I have a few questions about this section of the manuscript.

  1. Please indicate how many hams were used as experimental material in each of the four experimental groups. As can be seen from the description in the Material&Methods chapter, there were 10 animals in each group of pigs with different diets (line 100-101). 10 hams were obtained from each of both groups (line 116). How was the next step? These 10 hams were divided into two parts (n=5?), one of which was subjected to a normal smoking procedure and the other with a 50% reduction in application of smoke? Please specify.

Yes, 10 hams were used per feeding treatment: 10 from acorn-fed pigs (AH hams) + 10 from control-fed pigs (CH hams). Each group was then equally divided between the smoking treatments so that 5 AH and 5 CH were subjected to a standard smoking treatment (SH hams, n=10), while the other 5 AH and 5 CH were subjected to a 50% reduced smoking treatment (LSH hams, n=10).

The following description of experimental design is included in the text:

L131-135:

“The distribution of hams according to processing method and animal’s diet was a 2 x 2 factorial design, with half of the less smoked hams coming from control-fed pigs and the other half from acorn-fed pigs, and the same distribution was used for the standard smoking process (5 hams per each of 4 groups or 10 hams per effect of processing and diet).”

  1. Why were the samples frozen, since dry-cured hams are usually shelf-stable products? If this way possible biochemical changes were protected? How long were the hams stored frozen? Did the Authors take into account the fact that frozen storage may affect the mechanical properties, sensory quality of the product and initiate fat oxidation?

We agree with your comment that cured ham is indeed a shelf-stable product. However, as different analyses were planned in several facilities in different countries, we considered it best to freeze the samples until analysis in order to minimise possible changes (e.g. biochemical processes still going on) during storage and transport of the samples. We are aware that freezing could affect some of the properties mentioned, however we consider it was not a major effect as dry-cured ham is a long matured product with low aw.  As explained above, this was due to the circumstances. The samples were frozen for between 2 and 3 months depending on the analysis.

  1. How were samples cut for texture analysis using the TPA test? During the measurement, was the force directed perpendicular or parallel to the muscle fibers? Depending on the position of the test in relation to the compressing plate (or probe), we talk about a greater impact of the connective tissue (when the muscle fibers are arranged parallel to the base and perpendicular to the force) or a greater impact of the muscle fibers.

Thank you for this comment. We agree with you, although in comparison to fresh meat, in matured ham the position has less effect (as proteolysis affects both myofibrillar proteins and collagen).  In present study, for instrumental assessment of texture, the samples from SM and BF muscle were excised with a scalpel into parallelepipeds (dimensions of 20 mm × 20 mm × 15 mm for length, width and height, respectively). During the TPA measurements, samples were compressed perpendicular to the fibre bundle.

The explanation is added to the text at L193-196.

  1. Why did the Authors decide to compress the sample to 50% of its original height? Wouldn't it be better to have a destructive effect, i.e. 70-80%, in the case of products with significant gumminess and varying hardness? A much better correlation with the sensory evaluation is obtained when the sample is subjected to disintegration. Similar to what happens when it is crushed in the mouth.

We used the 50% compression based on our previous studies on dry-cured ham. From our experience, the higher compression may not be feasible for dry-cured ham, because, if the product is very dry (which often happens in SM) then hardness may be so big that it causes overload (no result and risk for apparatus).

Reviewer 2 Report

Comments and Suggestions for Authors

The manuscript by Karoyli et al. evaluated the effect of feed (acorn vs traditional) and processing conditions on the quality attributes of TP ham. The topic is novel, and the information could be very useful for increasing awareness, and consumers' interest in meat products prepared from this indigenous and slow-growing pig breed. Research in this aspect will ultimately help in improving the marketability and economic viability of TP. The hypothesis is clear and sound. The language is clear and easy to understand.

I have the following observations as-

·         L 22: Please check whether extinct should be replaced by endangered. As extinct meaning that this TP would no longer exist.

·         In Abstract: Please add the level of significance of the values. As the authors mentioned, the values are lower or higher, but readers would benefit more if they came to know whether these are significant or not.

·         L74-78- are used for emphasizing the hypothesis, so further detailing may further strengthen the hypothesis.

·         L103-104: control group received 3 kg feed, and treated TP received 4 kg; Please mention the reason (may be nutritional/ energy requirements etc)

·         L112: please mention the age of TP at slaughter

·         Results and discussion: appropriate and well supported by relevant references.

Author Response

Dear reviewer, We would like to thank you for your useful comments and suggestions for improving the manuscript. Below you will find our responses to your comments. All changes made have been incorporated into the revised manuscript by using the “Track Changes” tool for Word. Best regards yours, Danijel Karolyi Corresponding author

Reviewer 2

The manuscript by Karoyli et al. evaluated the effect of feed (acorn vs traditional) and processing conditions on the quality attributes of TP ham. The topic is novel, and the information could be very useful for increasing awareness, and consumers' interest in meat products prepared from this indigenous and slow-growing pig breed. Research in this aspect will ultimately help in improving the marketability and economic viability of TP. The hypothesis is clear and sound. The language is clear and easy to understand.

Thank you for your positive evaluation of our work and your support for research into local pig breeds.

I have the following observations as-

L 22: Please check whether extinct should be replaced by endangered. As extinct meaning that this TP would no longer exist.

Thank you for this comment. Due to the very small population size at the end of the 20th century, the Turopolje pig breed was included in the World Watch List for Domestic Animal Diversity among the animals threatened with extinction (Loftus and Scherf, 1993, World Watch List for Domestic Animal Diversity FAO Rome, Italy). In 1996, for example, only 12 sows and 3 boars were registered in the herd book of the Turopolje pig breed. Thanks to the initiated protection measures, the population of Turopolje pigs has slowly increased in recent years, but at that time the breed was truly threatened with extinction.

In Abstract: Please add the level of significance of the values. As the authors mentioned, the values are lower or higher, but readers would benefit more if they came to know whether these are significant or not.

The p values are added into the abstract.

L74-78- are used for emphasizing the hypothesis, so further detailing may further strengthen the hypothesis.

Thank you for this valuable comment. To further support the hypothesis, the following paragraph on processing innovations in TP products has been added to the introduction:

L78-82:

“In addition, the health-related innovations in traditional meat products, such as smoke reduction [14], appear to be feasible for better commercialization of this breed, as no significant differences in consumer acceptance were found between less smoked and standard TP dry-cured ham [17].”

 L103-104: control group received 3 kg feed, and treated TP received 4 kg; Please mention the reason (may be nutritional/ energy requirements etc)

The difference in the amount of feed given to the pigs in the control and experimental groups in the final fattening phase was due to the different energy content of the feed mixture and the acorn. Hence, to compensate for the lower metabolisable energy of the acorn compared to the feed mixture, the acorn-fed pigs received a higher daily amount of feed. The following explanation is added to the text:

L109-111:

“The higher daily amount of feed in the acorn-fed group was intended to compensate for the lower metabolisable energy of the acorn compared to the feed mixture.”  

L112: please mention the age of TP at slaughter

The age TP at slaughter is already implied in the text, as the final fattening began at an average age of 506 days (L103) and this phase lasted 46 days (L107). However, for the sake of clarity, the average age at slaughter (552 days) is added to the text (L118).

Results and discussion: appropriate and well supported by relevant references.

Thank you very much for this comment.

Reviewer 3 Report

Comments and Suggestions for Authors

This manuscript contains a very detailed work regarding dry-cured hams originating from TP breed. The effects of the diet (acorn supplementation or conventional) and ham processing method (standard or less smoked) on the physicochemical, textural traits and volatile and sensory profile were analyzed. The study is valuable, but a major point would be to discuss in some part that the less smoking process was planned on the basis of earlier studies which assure that microbial loads were controlled equally to the standard process.

Some minor comments may be responded.

L112, the average live-weight at slaughter is extremely low for an age of 506+46 days. It would be valuable that you explain if the animals were fed restrictedly until the start of the experiment or if the average daily gain of the pigs and/or the adult size is normally so small.

L125-126 it should be stated if it was a 2 x 2 factorial design with half of the less smoked hams were control and the other half had been acorn-fed, and the same for standard processed (5 hams per each of the 4 groups). In case of VOC, it should be reported that only two hams per group were evaluated. The reader would acknowledge to be informed in the first sentence of results that, since there was no interaction between processing method and diet, the single effects are reported below, except for VOC acids.

Table 4. the reader would like to know the results of the interaction on acid compounds (at least a summary in text without the need to move on the supplementary materials)

Figure 1. it must be highlighted that processing method effects on VOC seem more important than on the rest of the variables of the manuscript, and also if this are linked with sensory evaluation.

Table 5. it seems that sensory characteristics (taste, flavor and texture attributes) could be impaired in Biceps femoris by less smoking? discuss

L402 the lower processing losses in AH may also be attributed to higher SFA content of acorn-supplemented diet (and thereby the subcutaneous fat composition of hams?)

L513 which would be the estimated voluntary feed intake for a 95-kg TP pig? Does it limit to consume more than 2kg of acorns?

Author Response

Dear reviewer, We would like to thank you for your useful comments and suggestions for improving the manuscript. Below you will find our responses to your comments. All changes made have been incorporated into the revised manuscript by using the “Track Changes” tool for Word. Best regards yours, Danijel Karolyi Corresponding author

Reviewer 3

This manuscript contains a very detailed work regarding dry-cured hams originating from TP breed. The effects of the diet (acorn supplementation or conventional) and ham processing method (standard or less smoked) on the physicochemical, textural traits and volatile and sensory profile were analyzed. The study is valuable, but a major point would be to discuss in some part that the less smoking process was planned on the basis of earlier studies which assure that microbial loads were controlled equally to the standard process.

Thank you for your appreciation of our work.

The processing innovation in the form of reduced smoke application was selected based on a previous study with Croatian consumers, which showed that smoke reduction is the most accepted health-related innovation in traditional meat products (Karolyi, D.; Cerjak, M. The acceptance of health related innovations in traditional meat products by Croatian consumers. Poljoprivreda 2015, 21 (Suppl. 1), 228-231). This has already been mentioned in the discussion (L475-478), and we have now also referred to it in the introduction (L78-82).  

As far as the question of microbial loads is concerned, unfortunately no microbiological analyses were carried out in the present study. However, we did not find any significant differences in physicochemical parameters, such as aw and pH values between less smoked and standard hams, which could at least indirectly indicate similar microbial control in both processes. Accordingly, the discussion is amendment with the following paragraph:

L504-508:

“Regarding the antimicrobial effect of the smoking process, no microbiological analyses were performed in the present study. However, we found no significant differences in physicochemical parameters such as aw and pH values between LSH and SH, which could at least indirectly indicate similar microbial control in both processes.”

Some minor comments may be responded.

L112, the average live-weight at slaughter is extremely low for an age of 506+46 days. It would be valuable that you explain if the animals were fed restrictedly until the start of the experiment or if the average daily gain of the pigs and/or the adult size is normally so small.

Yes, we agree that the average live weight of the pigs at slaughter in the present study is very low compared to the usual live weight of pigs of similar age. The Turopolje pig is a local, unimproved pig breed that has never been selected for its growth potential and is therefore characterised by a much slower growth rate than modern pigs. This became particularly clear under the conditions of the present study, in which the pigs were kept extensively in a free-range system with free access to natural feed sources and some supplementary feed. In addition, the pigs used entered the first phase of the trial with an average age of 400 days and a live weight of 45 kg, which imply a restrictive condition until the start of the experiment, but this was beyond our control.

We add the information on starting age and weight of pigs in the text (L100-101).

L125-126 it should be stated if it was a 2 x 2 factorial design with half of the less smoked hams were control and the other half had been acorn-fed, and the same for standard processed (5 hams per each of the 4 groups). In case of VOC, it should be reported that only two hams per group were evaluated. The reader would acknowledge to be informed in the first sentence of results that, since there was no interaction between processing method and diet, the single effects are reported below, except for VOC acids.

Thank you very much for this comment. The following descriptions of experimental design and statistical analysis are included in the text, as suggested:

L131-135:

“The distribution of hams according to processing method and animal’s diet was a 2 x 2 factorial design, with half of the less smoked hams coming from control-fed pigs and the other half from acorn-fed pigs, and the same distribution was applied for the standard smoking process (5 hams per each of 4 groups or 10 hams per effect of processing and diet).”

L202-203:

“…(2 for each of the 4 groups of 4 per effect of processing and diet).”

L252-254:

“Since there was no interaction between processing method and diet, the single effects are reported below, with the exception of VOCs.”

L303:

“…and their interaction…”

Table 4. the reader would like to know the results of the interaction on acid compounds (at least a summary in text without the need to move on the supplementary materials)

The following sentence is added to the text, as suggested:

L336-339:

“In addition, total acids were significantly higher in the LSH of acorn-fed group than in the SH from both diets, while in the control-fed group, acid levels between smoking processes were similar and comparable to those observed in acorn diet counterparts.”   

Figure 1. it must be highlighted that processing method effects on VOC seem more important than on the rest of the variables of the manuscript, and also if this are linked with sensory evaluation.

Thank you for this comment. Figure 1. shows the effect sizes (Hedge’s g value) of animal’s diet and processing on main groups of VOCs, were g > 0.8 standard deviations indicating a large effect. The magnitude of the effect size of processing on VOCs is emphasised in L359-368. However, because Hedge’s g value was only calculated for VOCs, as an adjunct to ANOVA due to the small sample size for VOCs, it is not possible to comment on it in relation to other variables analysed. In any case, we have referred in the Discussion section to the effect of processing, i.e. the reduction of smoking, on VOCs and sensory evaluation (L514-522), and we hope you will find this this answer satisfactory.

Table 5. it seems that sensory characteristics (taste, flavor and texture attributes) could be impaired in Biceps femoris by less smoking? Discuss

Thank you for this observation. Indeed, the meat colour intensity tended to be impaired by decrease in smoking and this was corrected in the manuscript (L387).

However, for other sensory traits, such as, sourness and bitterness, the reduction in smoking appears to be beneficial, as these traits tended to be scored lower in LSH than in SH. Moreover, as shown in Table 5. and pointed out in the abstract, discussion and conclusions, the positive effect of reduced smoking was particularly evident in the perceived texture of the dry-cured ham.      

L402 the lower processing losses in AH may also be attributed to higher SFA content of acorn-supplemented diet (and thereby the subcutaneous fat composition of hams?)

Yes, we agree with your observation. The higher SFA content in the acorn diet could contribute to a more saturated FA profile of subcutaneous fat, which could also be one of the factors for the lower processing losses in AH. However, we did not analyse the FA profile of hams.    

L513 which would be the estimated voluntary feed intake for a 95-kg TP pig? Does it limit to consume more than 2kg of acorns?

We do not know exactly for TP what would be the capacity of voluntary feed intake. We based the diets so as to be iso-energetics and in line with theoretical estimation (3-4 times maintenance energetic needs). It is known from the literature (Čandek-Potokar et al. 2019., Analytic Review of Productive Performances of Local Pig Breeds, in European Local Pig Breeds - Diversity and Performance. A study of project TREASURE, IntechOpen, London, UK) that feed intake is higher in autochthonous breeds, and according to the mentioned literature, the highest intake reported was 4.7 kg (feed mixture) and 5.6 kg (acorns) for Iberian pig. In the present study, we were unable to offer more than 2 kg of acorns per pig per day due to limited resources.

Reviewer 4 Report

Comments and Suggestions for Authors

The review of the manuscript (ID: animals-2813177) entitled: “Effect of animal’s diet and processing method on quality traits of dry-cured ham produced from Turopolje pigs.”

Dear Authors,

with great interest I read the scientific article, which is correctly written and contributes significantly to the current state of knowledge information in terms of the effect of the diet with acorn supplementation and ham processing method (standard or less smoked) on the physicochemical and textural traits and volatile and sensory profile of dry-cured ham originating from Turopolje pigs breed.

This is especially important in the aspect of use by the traditional production system  local pig breeds such as Turopolje pig, local fatty pig breed that originated in the Turopolje region of central Croatia (with the latest census of 240 sows and 18 boars kept on 18 farms).

The introduction to the topic is short, but sufficient and explains the purpose of the research well. The description of the course of the experiment is clear and clearly described. A very detailed methodology allows other scientists to repeat or continue their research in this area.

Correctly selected methodologies and tools allowed the authors to present the results in an interesting way and to discuss them. The conclusions consistent with the evidence and arguments were presented and they are addressing the main goal.

Minor improvement,

Line 161 – add the pH meter calibration method, was there temperature compensation?

Line 146 – add the Chroma Meter calibration method and device date (° view angle, measurement or illumination area)

Line 184 – in the text is “chewiness (N)”, should be – “chewiness (N*mm)”

Point 2.6 - in the tables you use the significance level marked with the letter t-p<0.10. What is it, and on what basis did you estimate it? Please explain. In the methodology, you clearly wrote, "using Tukey's t-test at a significance level of 5%." So where does the 10% level come from?

Author Response

Dear reviewer, We would like to thank you for your useful comments and suggestions for improving the manuscript. Below you will find our responses to your comments. All changes made have been incorporated into the revised manuscript by using the “Track Changes” tool for Word. Best regards yours, Danijel Karolyi Corresponding author

Reviewer 4

The review of the manuscript (ID: animals-2813177) entitled: “Effect of animal’s diet and processing method on quality traits of dry-cured ham produced from Turopolje pigs.”

Dear Authors,

with great interest I read the scientific article, which is correctly written and contributes significantly to the current state of knowledge information in terms of the effect of the diet with acorn supplementation and ham processing method (standard or less smoked) on the physicochemical and textural traits and volatile and sensory profile of dry-cured ham originating from Turopolje pigs breed.

This is especially important in the aspect of use by the traditional production system  local pig breeds such as Turopolje pig, local fatty pig breed that originated in the Turopolje region of central Croatia (with the latest census of 240 sows and 18 boars kept on 18 farms).

The introduction to the topic is short, but sufficient and explains the purpose of the research well. The description of the course of the experiment is clear and clearly described. A very detailed methodology allows other scientists to repeat or continue their research in this area.

Correctly selected methodologies and tools allowed the authors to present the results in an interesting way and to discuss them. The conclusions consistent with the evidence and arguments were presented and they are addressing the main goal.

Thank you very much for the very positive evaluation of our work.

Minor improvement,

Line 161 – add the pH meter calibration method, was there temperature compensation?

The information on calibration of the pH meter is added into text:

L173-174:

“Two-point calibration with Testo buffer pH 4/7 set was performed before the measurements according to the manufacturer's instructions.”

The pH-electrode without integrated temperature sensor is used to determine the pH-value, while the medium temperature is measured with the NCT-sensor. During measurement and alignment, the medium temperature was automatically included in the process. 

Line 146 – add the Chroma Meter calibration method and device date (° view angle, measurement or illumination area)

The information on illuminant, measuring area and calibration of the Chroma Meter is added into text:

L182-184:

“…with ø 8mm measuring area. The CIE standard illuminant D65 is used as the light source for the measurement. Before the measurement, a calibration was carried out on the CR-A43 white calibration plate in D65 with Y = 93.5, x = 0.3156, y = 0.3319.”

Line 184 – in the text is “chewiness (N)”, should be – “chewiness (N*mm)”

Thank you for spotting this, it is corrected.

Point 2.6 - in the tables you use the significance level marked with the letter t-p<0.10. What is it, and on what basis did you estimate it? Please explain. In the methodology, you clearly wrote, "using Tukey's t-test at a significance level of 5%." So where does the 10% level come from?

We are sorry that we missed to explain this in subchapter 2.6. In addition to the statistical significance level of 5 % (i.e. p≤0.05) used, we have labelled the effects with a p-value that tend towards statistical significance (e.g. p<0.10) with t in the tables in order to facilitate the interpretation of the findings. The explanation is now added in the text:

L258-259:

“Effects tending towards statistical significance (p<0.10) were labelled with t in the tables.”

Round 2

Reviewer 1 Report

Comments and Suggestions for Authors

Thank you for sending replies to my comments. After taking into account the comments of all reviewers, the manuscript submitted for review became more valuable.

Reviewer 2 Report

Comments and Suggestions for Authors

Thank you for considering my observations and editing accordingly. The manuscript is improved and may be accepted for publication.

Reviewer 3 Report

Comments and Suggestions for Authors

The authors addressed correctly the comments

Reviewer 4 Report

Comments and Suggestions for Authors

Thank you, all is correct.